# Terpenoids from the Seeds of *Toona sinensis* and Their Ability to Attenuate High Glucose-Induced Oxidative Stress and Inflammation in Rat Glomerular Mesangial Cells

**DOI:** 10.3390/molecules27185784

**Published:** 2022-09-07

**Authors:** Ying Chen, Hong Gao, Xiaoxiao Liu, Jinyi Zhou, Yijin Jiang, Feng Wang, Rongshen Wang, Wanzhong Li

**Affiliations:** School of Pharmacy, Weifang Medical University, Weifang 261053, China

**Keywords:** *Toona sinensis* (A. Juss.) roem, acyclic diterpenoids, rat glomerular mesangial cells, oxidative stress, inflammation

## Abstract

*Toona sinensis* (A. Juss.) Roem is an edible medicinal plant that belongs to the genus *Toona* within the Meliaceae family. It has been confirmed to display a wide variety of biological activities. During our continuous search for active constituents from the seeds of *T. sinensis*, two new acyclic diterpenoids (**1**–**2**), together with five known limonoid-type triterpenoids (**3**–**7**), five known apotirucallane-type triterpenoids (**8**–**12**), and three known cycloartane-type triterpenoids (**13**–**15**), were isolated and characterized. Their structures were identified based on extensive spectroscopic experiments, including nuclear magnetic resonance (NMR), high-resolution electrospray ionization mass spectra (HR-ESI-MS), and electronic circular dichroism (ECD), as well as the comparison with those reported in the literature. We compared these findings to those reported in the literature. Compounds **5**, **8**, and **13**–**14** were isolated from the genus *Toona*, and compounds **11** and **15** were obtained from *T. sinensis* for the first time. The antidiabetic nephropathy effects of isolated compounds against high glucose-induced oxidative stress and inflammation in rat glomerular mesangial cells (GMCs) were assessed in vitro. The results showed that new compounds **1** and **2** could significantly increase the levels of Nrf-2/HO-1 and reduce the levels of NF-κB, TNF-*α*, and IL-6 at concentrations of 30 μM. These results suggest that compounds **1** and **2** might prevent the occurrence and development of diabetic nephropathy (DN) and facilitate the research and development of new antioxidant and anti-inflammatory drugs suitable for the prevention and treatment of DN.

## 1. Introduction

Diabetic nephropathy (DN) is a chronic microvascular complication with a high incidence and mortality. It is the principal cause of end-stage renal disease worldwide [1]. The pathogenesis of DN is complicated and obscure, and there is currently no efficient way to alleviate DN. Oxidative stress and inflammation are the major causes of DN progression [2]. The release of reactive oxygen species (ROS) and the accumulation of inflammatory mediators can be very high in rat glomerular mesangial cells (GMCs) exposed to high glucose (HG) [3]. Continuing to search for a prospective inhibitor with antioxidative and anti-inflammatory properties may reveal an effective way to halt the progression of DN.

Nuclear factor erythroid 2-related factor 2 (Nrf2) has been proven to prevent ROS-induced oxidative stress injury and is a master transcriptional regulator of genes that encode antioxidative factors, such as heme oxygenase-1 (HO-1) [4]. Nuclear factor kappa-B (NF-κB) is an important transcription factor. It plays a vital role in the inflammatory response and mediates the expression of inflammatory cytokines involved in DN [5]. TNF-α and IL-6 are pro-inflammatory cytokines that play an important role in inflammatory responses, lipid metabolism, and insulin resistance. It was reported that TNF-α and IL-6 were associated with increased oxidative stress and inflammation in DN. NF-κB, TNF-α, and IL-6 are activated in renal tissue, and they have been reported as important biomarkers of DN [6,7]. The Nrf2/NF-κB pathway could be used to modulate oxidative stress and inflammation and so affect DN [8,9]. Hence, the Nrf2/NF-κB pathway may be a suitable target for the treatment of DN.

*Toona sinensis* (A. Juss.) Roem, a medicinal and edible plant that belongs to the genus *Toona* within the Meliaceae family, is mainly distributed in Asia and Oceania [10,11]. The tender leaves and buds are not only eaten as vegetables but also possess considerable value as a folk medicine in the treatment of heliosis, vomiting, dysentery, enteritis, and itchiness [12]. The roots can be used as astringents and the stems as correctives [13]. *T. sinensis* is rich in a variety of active components, including terpenoids, flavonoids, lignans, and phenols, and it exerts antioxidant, anti-inflammatory, anti-diabetic, and anti-tumor effects [14,15]. At present, studies have shown that the chemical constituents of *T. sinensis* are mainly concentrated on the leaves and bark, and there have been a few reports on the seeds. The results showed that the seeds mainly contained terpenoids, flavonoids, and phenols [13,16,17]. Further exploration may identify the active constituents of *T. sinensis* seeds that improve oxidative stress and inflammation in DN.

A phytochemical study on the seeds of *T. sinensis* led to the isolation and identification of two new acyclic diterpenoids (**1**, **2**), together with five known limonoid-type triterpenoids (**3**–**7**), five known apotirucallane-type triterpenoids (**8**–**12**), and three known cycloartane-type triterpenoids (**13**–**15**), which were elucidated based on extensive spectral data. The antidiabetic nephropathy effects of isolated compounds against HG-induced oxidative stress and inflammation in GMCs were assessed in vitro. The new compounds **1** and **2** could significantly increase the levels of Nrf-2/HO-1 and reduce levels of NF-κB, TNF-α, and IL-6 at concentrations of 30 μM. Herein, we report the isolation and structure identification of terpenoids (**1**–**15**) (Figure 1) from the *T. sinensis* seeds together with the protective effect against DN through the activation of Nrf2/HO-1 and inhibition of NF-κB pathways.

## 2. Results and Discussion

Compound **1** was isolated as a white gum and had a molecular formula of C_20_H_36_O_4_ based on its high-resolution electrospray ionization mass spectra (HR-ESI-MS) (Appendix A), which showed a peak at *m*/*z* 341.26270 [M+H]^+^ (calcd. 341.26864), corresponding to three degrees of unsaturation. The ^1^H NMR spectrum (Appendix A) of **1** exhibited resonances assignable to three olefinic proton signals at *δ*_H_ 5.45 (1H, t, *J* = 6.7 Hz, H-10), 5.39 (1H, t, *J* = 6.8 Hz, H-2), and 5.09 (1H, t, *J* = 6.1 Hz, H-6); two oxygenated methine proton signals at *δ*_H_ 4.52 (1H, m, H-12) and 3.98 (1H, m, H-14); and five methyl signals at *δ*_H_ 1.66 (3H, s, H-17), 1.58 (6H, s, H-18, 19), 1.26 (3H, s, H-20), and 1.23 (3H, s, H-16). There were protons for the oxygenated methylene at *δ*_H_ 4.14 (2H, d, *J* = 6.8 Hz, H-1). Combined with the distortionless enhancement by polarization transfer (DEPT) 135° and HMQC data, 20 carbon signals in the ^13^C NMR spectrum (Appendix A) were assigned to be three olefinic quaternary carbons at *δ*_C_ 139.4 (C-3), 135.1 (C-7), and 134.6 (C-11); three olefinic methines at *δ*_C_ 126.8 (C-10), 124.3 (C-6), and 123.8 (C-2); one oxygenated quaternary carbon at *δ*_C_ 83.0 (C-15); two oxygenated methines at *δ*_C_ 81.9 (C-12) and 78.6 (C-14); one oxygenated methylene at *δ*_C_ 59.5 (C-1); five methylenes at *δ*_C_ 39.6 (C-4), 39.3 (C-8), 39.1 (C-13), 26.2 (C-5), and 26.1 (C-9); and five methyls at *δ*_C_ 28.0 (C-16), 21.7 (C-20), 16.4 (C-17), 16.1 (C-18), and 11.5 (C-19) (Table 1). All the above spectroscopic data, as well as the degrees of unsaturation, indicated that **1** was an acyclic diterpenoid. Analysis of the ^1^H-^1^H COSY and HMBC data of **1** led to the successful assignment of its planar structure as a diterpenoid (Figure 2). The ^1^H-^1^H COSY correlations of H_2_-1/H-2, H_2_-4/H_2_-5/H-6 and H_2_-8/H_2_-9/H-10, together with the HMBC correlations from H_2_-1 to C-2, C-3, from H_2_-5 to C-6, C-7, and from H_2_-9 to C-10, C-11, suggested the presence of three double bonds respectively assigned as Δ2, Δ6, Δ10. Two oxygenated methines were respectively assigned to C-12 and C-14 by ^1^H-^1^H COSY correlations of H-12/H_2_-13/H-14 and HMBC correlations from H-12 to C-13, C-19, from H-14 to C-12, C-15, and from H_3_-16, H_3_-20 to C-14. The key HMBC correlations from H_3_-16 to C-14, C-15, from H_3_-18 to C-6, C-7, C-8, from H_3_-19 to C-10, C-11, C-12, and from H_3_-20 to C-14, C-15 suggested that four methyl groups were located at the C-16, C-18, C-19, and C-20 positions, respectively. Comparing the NMR data of **1** with those of 3-(hydroxymethyl)-1, 12, 14, 15-tetrahydroxy-7, 11, 15, 15-tetramethyl-2, 6, 10-hexadecatriene [18] revealed that their chemical shifts were similar and the structural difference between **1** and the known compound 3-(hydroxymethyl)-1, 12, 14, 15-tetrahydroxy-7, 11, 15, 15-tetramethyl-2, 6, 10-hexadecatriene was that the hydroxymethyl group in the known compound was replaced by the methyl group in **1**. The key HMBC correlations from H_3_-17 to C-2, C-3, C-4 indicated that the methyl group was located at the C-17 position (Figure 2). The assignment of the double bond configurations at Δ2, Δ6, and Δ10 with *E*-geometry was secured by prominent spatial NOESY correlations observed between H-2/H_2_-4 and H_2_-1/H_3_-17 (for Δ2), H-6/H_2_-8 and H_2_-5/H_3_-18 (for Δ6), as well as H-10/H-12 and H_2_-9/H_3_-19 (for Δ10). The relative configuration of **1** was substantiated by the DP4+ probability analysis. The results indicated that 12-OH and 14-OH assume a *trans* orientation (Appendix A). The absolute configuration of **1** was defined by an electronic circular dichroism (ECD) experiment, and the experimental ECD spectrum of **1** was consistent with the calculated ECD of (12*S*, 14*S*)-**1** (Figure 3). Thus, the absolute configuration of **1** was established as shown in Figure 1. From the above evidence, compound **1** was finally determined to be 1, 12, 14, 15-tetrahydroxy-3, 7, 11, 15, 15-pentamethyl-2, 6, 10-hexadecatriene.

Compound **2** was obtained as a white gum and gave a molecular formula of C_22_H_38_O_5_ according to its HR-ESI-MS (Appendix A) with a peak at *m*/*z* 405.26089 [M+Na]^+^ (calcd. 405.26115), corresponding to four degrees of unsaturation. Analysis of the NMR (Appendix A) data of **2** (Table 1) suggested that its structure closely resembled that of **1**, with these compounds sharing an identical carbon scaffold and substitution patterns, except the existence of an AcO group at C-1 (*δ*_C_ 171.4, 1-O*CO*CH_3_; *δ*_H_ 2.04, *δ*_C_ 21.2, 1-OCO*CH_3_*) in **2** rather than a hydroxy in **1**, as confirmed by the key HMBC correlations from 1-OCO*CH_3_* (*δ*_H_ 2.04) to 1-O*CO*CH_3_ (*δ*_C_ 171.4) and H-1 (*δ*_H_ 4.57) to 1-O*CO*CH_3_ (*δ*_C_ 171.4) (Figure 2). The configuration of **2** was characterized by the DP4+ probability analysis and ECD data. The *trans* orientation of H-12 and H-14 was established by the DP4+ probability analysis (Appendix A). The calculated ECD spectrum of (12*R*, 14*R*)-**2** well matched the corresponding experimental ECD curve (Figure 3). This resulted in the assignment of the absolute configuration of **2** as shown in Figure 1. As described above, compound **2** was consequently constructed as 1-*O*-acetyl-12, 14, 15-trihydroxy-3, 7, 11, 15, 15-pentamethyl-2, 6, 10-hexadecatriene.

Compared with the corresponding spectroscopic data in the literature, 13 known compounds were identified as gedunin (**3**) [19], 6*α*-hydroxygedunin (**4**) [20], 11*β*-hydroxygedunin (**5**) [21], 7-deacetoxy-7*α*,11*α*-dihydroxygedunin (**6**) [21], 7*α*-obacunyl acetate (**7**) [22], protoxylocarpin G (**8**) [23], 21*α*-methylmelianodiol (**9**) [24], 21*α*,25-dimethylmelianodiol (**10**) [25], hispidone (**11**) [26], bourjotinolone A (**12**) [27], 3*β*,25-dihydroxy-tirucalla-7,23-diene (**13**) [22], 3*β*,23-dihydroxy-tirucalla-7,24-diene (**14) [28]**, and 24,25-epoxy-3*β*,23-dihydroxy-7-tirucallene (**15**) [22].

The cytotoxicity of compounds isolated from the *T. sinensis* seeds was measured through a 3-(4, 5-dimethylthiazol-2-yl)-2, 5-diphenyl tetrazolium (MTT) assay. As shown in Appendix A, compounds **7**, **8**, **13**, and **15** exhibited significantly more cytotoxicity than the normal group (NG) (*p* < 0.01), and compounds **1**–**6**, **9**–**12**, and **14** (80 μM) showed no cytotoxicity against GMCs. To further explore the effects of different concentrations of compounds (**1**–**6**, **9**–**12**, and **14**) on GMC proliferation, we investigated the cell proliferation induced by HG though MTT assay. As indicated in Figure 4, there was significantly more GMC proliferation in the HG group than in the NG (*p* < 0.01). However, the increase could be reversed by epalrestat (EPA), and the effect of proliferation was dramatically inhibited after treatment with compounds **2**, **4**, **9**–**11**, and **14** in a dose-dependent manner.

Oxidative stress caused by HG plays a dominant role in the progression of DN. The Nrf2 pathway is an important defense system against oxidative stress. It plays an antioxidative stress role through the up-regulation of HO-1 and other antioxidant genes. The antioxidant effects were preliminarily evaluated based on the up-regulation of Nrf2/HO-1 in HG-stimulated GMCs. As shown in Figure 5, there was more Nrf2/HO-1 expression by GMCs increased in the HG group than by those in the NG group. In addition, Nrf2/HO-1 expression was further enhanced after incubation with compounds **1**, **2**, **4**, **6**, and **14** for 48 h. Taken together, these results proved that compounds **1**, **2**, **4**, **6**, and **14** can alleviate oxidative stress and are associated with the activation of the Nrf2/HO-1 pathway.

Increased oxidative stress is associated with the activation of inflammation. NF-κB is an important transcription factor that can regulate the expression of various inflammatory factors, such as TNF-*α* and IL-6. As shown in Figure 6, there was significantly more expression of inflammatory cytokines in the HG group than in the NG. Nevertheless, we found that HG-induced levels of NF-κB, IL-6, and TNF-*α* were suppressed by E*p*A and by compounds **1**, **2**, **4**, and **14** in a dose-dependent manner. However, compound **6** exhibited the same effect on NF-κB and TNF-*α* factors, but the inhibition of IL-6 was not dose-dependent. In short, compounds **1**, **2**, **4**, and **6** showed strong inhibitory activities at doses below 50 μM (*p* < 0.05 or *p* < 0.01), indicating that compounds **1**, **2**, **4**, and **6** suppressed the activation of the NF-κB pathway to block the progression of DN.

A preliminary structure–activity relationship was established, indicating that new acyclic diterpenoids (**1** and **2**) exerted significant antidiabetic nephropathy through the activation of Nrf2/HO-1 and the inhibition of NF-κB pathways with respect to HG-induced GMCs. Compounds **1** and **2** exhibited similar activities at concentrations of 30 μM (*p* < 0.05 or *p* < 0.01), perhaps because the functional group of C-1 or stereochemistry of C-12, 14 could not affect the strength of the regulatory activity. These data are reported as antidiabetic nephropathy agents herein for the first time.

## 3. Materials and Methods

### 3.1. Plant Material

The *T. sinensis* seeds were collected by the Jinan Shengke Technology Company (Jinan, China) and identified by Prof. Chongmei Xu. A voucher specimen (voucher number: WF–YXY–TSS1507) was deposited at the Pharmacognosy Laboratory of the School of Pharmacy, Weifang Medical University.

### 3.2. General Experimental Procedures

HR-ESI-MS were obtained using a Bruker microsoft time-of-flight QII mass spectrometer (Bruker Daltonics, Fremont, CA, USA). The NMR spectra were recorded using a Bruker AV 600/400 MHz spectrometer (Bruker, Fällanden, Switzerland). Optical rotation was measured using a Rudolph Autopol I automatic polarimeter (Rudolph Research Analytical, Hackettstown, NJ, USA). Column chromatography was performed using silica gel (200–300 mesh) and Sephadex LH-20 (Shanghai Yuanye Biological Technology Co., Ltd., Shanghai, China), and Lichroprep RP-18 gel (40–60 µm) was purchased from Merck KGaA (Darmstadt, Germany). Thin-layer chromatography (TLC) was performed with precoated silica gel GF 254 glass plates (100 mm × 200 mm, Branch of Qingdao Haiyang Chemical Co., Ltd.). All other chemicals and solvents were of analytical grade and used without further purification.

### 3.3. Extraction, Isolation and Purification

Dried *T. sinensis* seeds (55 kg) were extracted with 55%, 75%, and 95% ethanol (500 L × 3 times) with heating reflux for 2.5, 2, and 1.5 h, respectively [12,29,30]. The filtrate was combined and concentrated under vacuum to obtain a crude extract (5.2 kg), then sequentially partitioned with petroleum ether, EtOAc, and *n*-BuOH. The EtOAc extract (528.8 g) was purified by silica gel column chromatography and eluted with a gradient of petroleum ether:EtOAc (from 30:1 to 1:1, *v*/*v*) and CH_2_Cl_2_:MeOH (from 5:1 to 1:1, *v*/*v*) to obtain 9 fractions (fr. A–I). Fr. C (14.5 g) was further isolated by ODS and eluted with MeOH-H_2_O (from 40% to 70%) to obtain 13 subfractions (fr. C1–C13). Fr. C5 (2.6 g) was purified by silica gel column chromatography and eluted with a gradient of petroleum ether:EtOAc (from 20:1 to 10:1, *v*/*v*) to obtain 9 fractions (fr. C5.1–C5.9). Fr. C5.2 was separated using MPLC (C_18_, 40 g, 40-60 μm) and eluted with MeOH-H_2_O (from 80% to 100%) to yield compound **13** (28.6 mg). Fr. C5.3 was subjected to ODS and eluted with MeOH-H_2_O (from 80% to 100%) to obtain compound **14** (54.7 mg). Fr. C6 (3.2 g) was further chromatographically divided into 10 fractions (fr. C6.1–C6.10) using silica gel and eluted with CH_2_Cl_2_-MeOH (from 100:0 to 50:1, *v*/*v*). Compound **8** (22.4 mg) was obtained from fr. C6.1 through ODS eluted with MeOH-H_2_O (from 75% to 95%). Fr. C6.3 was isolated using ODS and eluted with MeOH-H_2_O (from 60% to 80%) to obtain compound **2** (14.6 mg). Fr. C6.5 was separated by ODS and eluted with MeOH-H_2_O (from 70% to 85%) to acquire compound **12** (21.6 mg). Compound **15** (33.6 mg) was purified from fraction C6.10 by ODS though MeOH-H_2_O (from 80% to 95%). Fr. C7 (6.7 g) was chromatographed with a gradient of CH_2_Cl_2_:MeOH (from 100:0 to 50:1, *v*/*v*) to attain 5 fractions (fr. C7.1–C7.5). Fr. C7.2 was subjected to silica gel with a gradient of petroleum ether:EtOAc (from 18:1 to 6:1, *v*/*v*) to obtain 4 fractions (fr. C7.2.1-C7.2.4). Fr. C7.2.1 was separated by ODS and eluted with MeOH-H_2_O (from 55% to 80%) to obtain compound **9** (46.2 mg). Compound **10** (19.3 mg) was purified from fraction C7.2.3 by silica gel column chromatography with a gradient of petroleum ether:EtOAc (from 15:1 to 8:1, *v*/*v*) and ODS eluted with a gradient of MeOH-H_2_O (from 75% to 100%). Fr. C7.3 was separated by ODS and eluted with MeOH-H_2_O (from 65% to 85%) to obtain compound **11** (19.7 mg). Fr. E (18.9 g) was further segmented into 10 fractions (fr. E1–E10) by MPLC (C_18_, 80 g, 40-60 μm) and eluted with MeOH-H_2_O (from 50% to 100%). Fr. E5 (3.2 g) was parted using silica gel column chromatography with a gradient of petroleum ether:EtOAc (from 7:1 to 4:1, *v*/*v*) to acquire compound **4** (41.4 mg). Fr. E6 (2.6 g) was chromatographed by silica gel using CH_2_Cl_2_:MeOH as a gradient eluent (from 100:1 to 50:1, *v*/*v*) to attain compounds **3** (23.6 mg) and **6** (38.7 mg). Fr. E7 (2.8 g) was isolated by silical gel column chromatography using CH_2_Cl_2_:MeOH (from 100:0 to 85:1, *v*/*v*) to obtain compounds **5** (17.6 mg) and **7** (27.7 mg). Fr. E8 (1.8 g) was subjected to a silica gel column chromatography gradient eluted with petroleum ether:EtOAc (from 7:1 to 4:1, *v*/*v*) and ODS using MeOH-H_2_O (from 50% to 100%) to obtain compound **1** (15.2 mg).

1, 12, 14, 15-tetrahydroxy-3, 7, 11, 15, 15-pentamethyl-2, 6, 10-hexadecatriene (**1**): white gum; C_20_H_36_O_4_; [*α*${]}_{D}^{21}$−1.07 (*c* 1.05, MeOH); HR-ESI-MS (positive ion mode) *m*/*z* 341.26270 [M+H]^+^ (calcd. for C_20_H_37_O_4_, 341.26864); ^1^H NMR (CDCl_3_, 600 MHz); and ^13^C NMR (CDCl_3_, 150 MHz), which were unambiguously assigned by DEPT 135°, ^1^H–^1^H COSY, HMQC, HMBC, and NOESY experiments (Appendix A) (see Table 1).

1-*O*-acetyl-12, 14, 15-trihydroxy-3, 7, 11, 15, 15-pentamethyl-2, 6, 10-hexadecatriene (**2**): white gum; C_22_H_38_O_5_; [*α*${]}_{D}^{21}$+5.38 (*c* 0.35, MeOH); HR-ESI–MS (positive ion mode) *m*/*z* 405.26089 [M+Na]^+^ (calcd. for C_22_H_38_O_5_Na, 405.26115); ^1^H NMR (CDCl_3_, 600 MHz); and ^13^C NMR (CDCl_3_, 150 MHz), which were unambiguously assigned by DEPT 135°, ^1^H–^1^H COSY, HMQC, HMBC, and NOESY experiments (Appendix A) (see Table 1).

### 3.4. Cytotoxicity Assay

GMCs (Institute of Nanjing Jiancheng) were incubated in a 5% CO_2_ atmosphere at 37 °C and cultured in 5.6 mmol/L glucose DMEM, which contained 10% foetal bovine serum and a 1% penicillin streptomycin solution. Cytotoxic activity was tested with MTT. The cells were plated on 96-well plates at 4 × 10^3^ cells per well, then incubated at 37 °C for 24 h. Then, the cells were cultured in NG and were co-treated with or without compounds (**1**–**15**) at 80 μM and incubated for 48 h. In addition, 10 μL MTT was added to each well. After 4 h, DMSO was added to dissolve formazan crystals. The absorbance was measured at 490 nm.

### 3.5. Cell Proliferation Assay

MTT assay was used to detect the effect of compounds without toxicity on cell proliferation. Cells were plated into 96-well plates and incubated for 24 h at 37 °C. The cells were divided into 8 groups: NG, 5.6 mmol/L glucose; HG group, 25 mmol/L glucose; HG+EPA (10 μM) group; and HG+compounds at different concentrations (5, 10, 20, 40, and 80 µM) group.

### 3.6. Elisa Assay

Before the experiment, the treated proteins and supernatants were stored at −80 °C. The levels of Nrf2, HO-1, NF-κB, TNF-*α*, and IL-6 in the cell supernatants and proteins were detected using commercially available Elisa kits (Nanjing Jiancheng Biology Engineering Institute, Nanjing, China) according to the manufacturer′s instructions.

### 3.7. Statistical Analysis

All data are presented as the means ± SD from 3 replicates. SPSS 22.0 software was used for one-way ANOVA of multiple groups of data. It was considered a significant difference when the *p* value was less than 0.05.

## 4. Conclusions

In this study, two new diterpenoids and thirteen known triterpenoids were separated from *T. sinensis* seeds. Compounds **1**, **2**, **4**, **6**, and **14** were found to increase the expression of Nrf2/HO-1 and decrease the levels of NF-κB, TNF-α, and IL-6, which indicated that the bioactive terpenoids of *T. sinensis* seeds could activate the Nrf2/HO-1 pathway and suppress the NF-κB pathway to ameliorate oxidative stress and inflammation, further preventing and reducing the occurrence of DN. This study indicated that the reasonable consumption of *T. sinensis* seeds might be an effective way to halt DN progression.

## Figures and Tables

**Figure 1 molecules-27-05784-f001:**
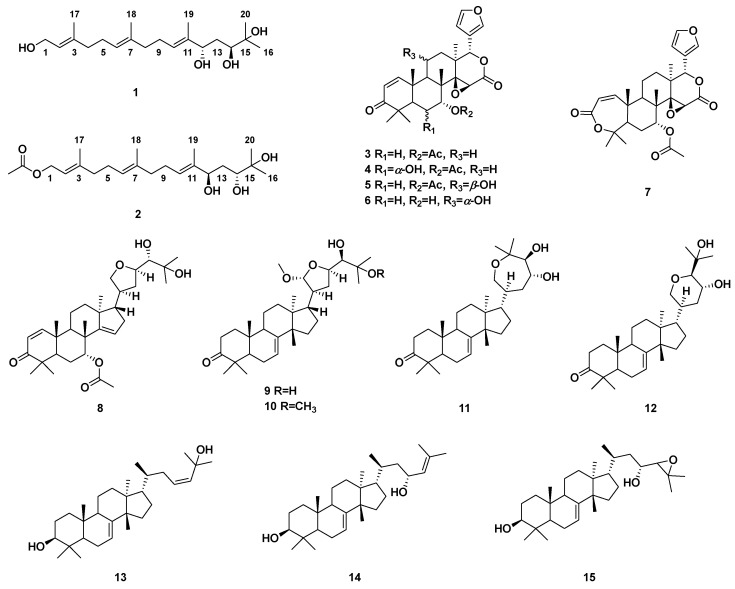
Structures of compounds **1**–**15** from the seeds of *T. sinensis*.

**Figure 2 molecules-27-05784-f002:**
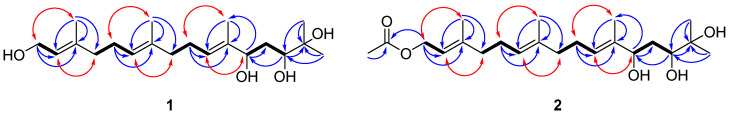
Key ^1^H-^1^H COSY (bold lines), HMBC (blue arrows), and NOESY (red arrows) correlations of compounds **1** and **2**.

**Figure 3 molecules-27-05784-f003:**
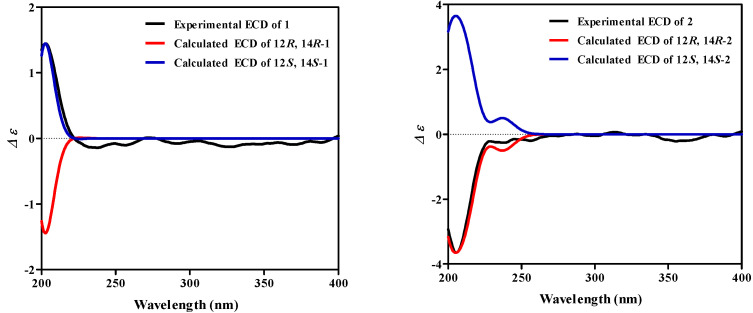
Calculated and experimental ECD spectra of compounds **1**–**2**.

**Figure 4 molecules-27-05784-f004:**
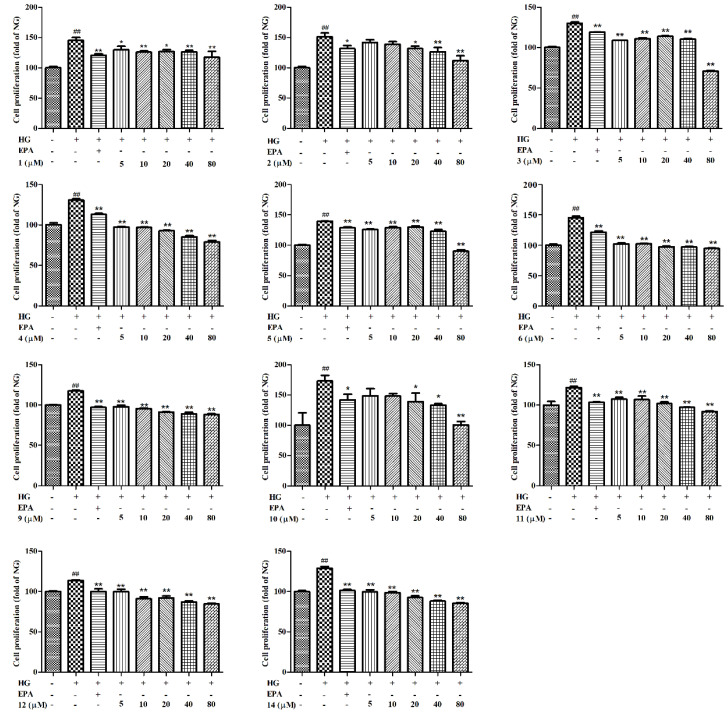
Cell proliferation in GMCs was detected with MTT assay with various concentrations of **1**–**6**, **9**–**12**, and **14** (5, 10, 20, 40, 80 μM) for 48 h. Values are expressed as mean ± SD of three independent experiments, with ^##^
*p* < 0.05 relative to the NG and * *p* < 0.05 or ** *p* < 0.01 relative to the HG group.

**Figure 5 molecules-27-05784-f005:**
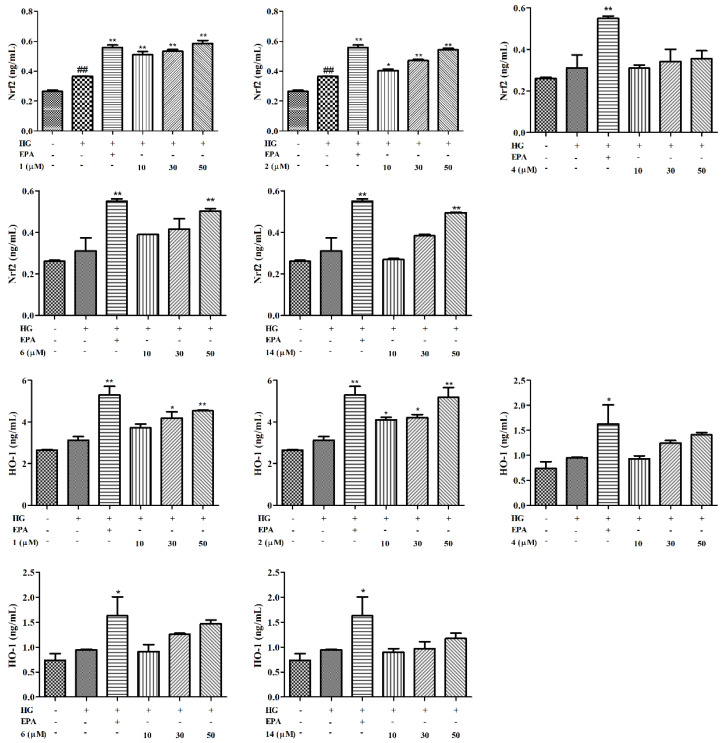
Effects of compounds on oxidative stress in GMCs. The GMCs were treated with or without compounds **1**, **2**, **4**, **6**, **14** at concentrations of 10, 30, and 50 µM in HG for 48 h. The expressions of Nrf2/HO-1 were analyzed using commercially available Elisa kits. Values are expressed as mean ± SD of three independent experiments, with ^##^
*p* < 0.05 relative to the NG and * *p* < 0.05 or ** *p* < 0.01 relative to the HG group.

**Figure 6 molecules-27-05784-f006:**
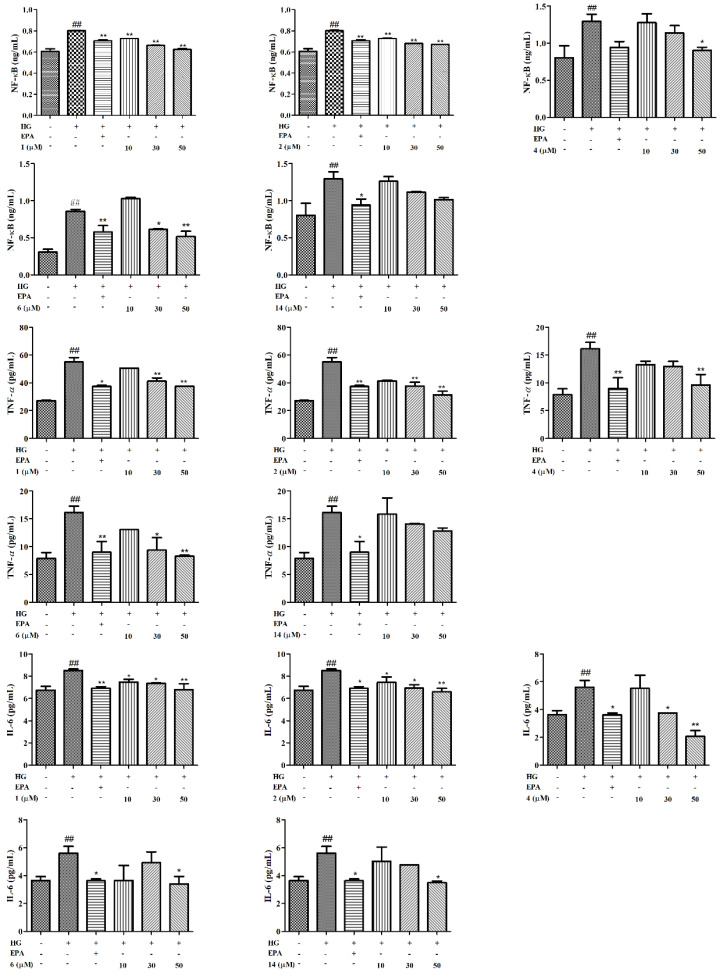
Effects of compounds on inflammation in GMCs. The GMCs were treated with or without compounds **1**, **2**, **4**, **6**, **14** at concentrations of 10, 30, and 50 µM in HG for 48 h. The expressions of NF-κB, TNF-*α*, and IL-6 were analyzed using commercially available Elisa kits. Values are expressed as mean ± SD of three independent experiments, with ^##^
*p* < 0.05 relative to the NG and * *p* < 0.05 or ** *p* < 0.01 relative to the HG group.

**Table 1 molecules-27-05784-t001:** ^1^H-NMR (600 MHz, CDCl_3_) and ^13^C-NMR (150 MHz, CDCl_3_) chemical shifts of compounds **1** and **2** (*δ* in ppm).

NO.	1		2	
	*δ*_H_ (*J* in Hz)	*δ* _C_	*δ*_H_ (*J* in Hz)	*δ* _C_
1	4.14 (2H, d, 6.8)	59.5	4.57 (2H, d, 7.1)	61.5
2	5.39 (1H, t, 6.8)	123.8	5.32 (1H, t, 7.1)	118.4
3	-	139.4	-	142.3
4	2.05 (2H, m)	39.6	2.02 (2H, m)	39.5
5	2.11 (2H, m)	26.2	2.15 (2H, m)	26.4
6	5.09 (1H, t, 6.1)	124.3	5.07 (1H, t, 6.3)	124.4
7	-	135.1	-	135.1
8	2.03 (2H, m)	39.3	2.02 (2H, m)	39.5
9	2.11 (2H, m)	26.1	2.08 (2H, m)	26.2
10	5.45 (1H, t, 6.7)	126.8	5.21 (1H, t, 6.8)	129.5
11	-	134.6	-	131.8
12	4.52 (1H, m)	81.9	3.68 (1H, t, 7.9)	69.4
13	2.01 (1H, m); 1.91 (1H, m)	39.1	2.61 (1H, m); 2.05 (1H, m)	45.1
14	3.98 (1H, m)	78.6	3.29 (1H, m)	79.0
15	-	83.0	-	73.1
16	1.23 (3H, s)	28.0	1.25 (3H, s)	26.3
17	1.66 (3H, s)	16.4	1.68 (3H, s)	16.6
18	1.58 (3H, s)	16.1	1.58 (3H, s)	15.9
19	1.58 (3H, s)	11.5	1.65 (3H, s)	16.1
20	1.26 (3H, s)	21.7	1.29 (3H, s)	25.5
1-O*CO*CH_3_	-	-	-	171.4
1-OCO*CH_3_*	-	-	2.04 (3H, s)	21.2

## Data Availability

Data are contained within the article.

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
