# Peer review of "Terpenoids from the Seeds of Toona sinensis and Their Ability to Attenuate High Glucose-Induced Oxidative Stress and Inflammation in Rat Glomerular Mesangial Cells"

_molecules, 2022, doi:10.3390/molecules27185784_

Round 1

Reviewer 1 Report

1.  It is not possible to determine the relative configuration of the flexible chain according to NOESY, it is recommended to do a propylene-fork reaction or use the single crystal results to determine the configuration.

2. Please check the compound coupling constants carefully.

3. The compound is not sufficiently novel and the conformational issues are not resolved.

4. Please put pictures of compounds together, one picture per compound is not recommended.

Reviewer 2 Report

This manuscript describes the isolation of terpenes from the seeds of Toona sinensis and evaluation of their antidiabetic nephropathy effects. The discovery of structurally diverse terpenes including two new ones make their work interesting for readers. However, the structure elucidation of the new compounds was not solid, especially the configurations. Revisions are required if the article is considered for publication.

Question 1: What roles do TNF-α and IL-6 play in the treatment of DN? Please specify it in the Introduction section.

Question 2: Is this the first phytochemical study of Toona sinensis seeds? Please give a detailed introduction of the chemical work on this species.

Question 3: The structural difference between 1 and the known compound 3-(hydroxymethyl)-1,12,14,15-tetrahydroxy-7,11,15,15-tetramethyl-2,6,10-hexadecatriene was that the hydroxymethyl group in the known compound was replaced by the methyl group in 1. Please correct it in the manuscript.

Question 4: Because the single bond can rotate freely, it is risky to determine the relative configurations of the chiral centers of the acyclic compounds 1 and 2. One way to solve this problem is to make the acetonate derivative, then use the NOESY spectrum to determine the relative configurations of the chiral centers in the cyclohexane ring. And don’t forget to elucidate the configuration of the three double bonds.

Question 5: Why use different concentrations of ethanol and reflux times for extraction? Was compound 2 an artifact? EtOAc was used for the partition and column chromatography.

1. P1L16:  ‘…based on extensive spectroscopic experiments, and they were found to contain nuclear magnetic resonance (NMR), high-resolution electrospray ionisation mass spectra (HR-ESI-MS), and electronic circular dichroism (ECD). We compared these findings to those reported in the literature.’ → ‘…based on extensive spectroscopic experiments including nuclear magnetic resonance (NMR), high-resolution electrospray ionisation mass spectra (HR-ESI-MS), and electronic circular dichroism (ECD), as well as the comparison with those reported in the literature.’

2. P3L78:  There are protons for the oxygenated methylene.

3. P3L93:  ‘…three double bonds respectively assigned to C-2, 3, C-6, 7, C-10, 11.’ → ‘…three double bonds respectively assigned as Δ2, Δ6 and Δ10.’

4. P3L101: The structural difference between 1 and the known compound 3-(hydroxymethyl)-1,12,14,15-tetrahydroxy-7,11,15,15-tetramethyl-2,6,10-hexadecatriene was that the hydroxymethyl group in the known compound was replaced by the methyl group. Please correct it in the manuscript.

5. P4L115:  ‘an peak’ → ‘a peak’

6. P5L141:  ‘The cytotoxicity of compounds isolated from the T. sinensis seeds was measured 141 through a 3-(4, 5-dimethylthiazol-2-yl)-2, 5-diphenyl tetrazolium (MTT) assay against 142 GMCs to confirm that the inhibitory effects of these compounds were not caused by their 143 cytotoxicity.’ This sentence was obscure.

Round 2

Reviewer 1 Report

The author cannot or does not want to address some of the core issues I mentioned earlier.

Reviewer 2 Report

The resubmitted manuscript has been clearly corrected/improved by the authors, and could be accepted.